# Study protocol for a single-blind, parallel-group, randomised, controlled non-inferiority trial of 4-day intensive versus standard cognitive behavioural therapy for adults with obsessive–compulsive disorder

Ekaterina Ivanova ,[1] Robin Fondberg,[2] Oskar Flygare ,[1] Max Sannemalm,[1] Sofia Asplund,[3] Sofia Dahlén,[3] Filipa Sampaio ,[4] Erik Andersson ,[5] David Mataix-Cols,[1,6] Volen Z Ivanov,[1] Christian Rück [1]

EI and RF contributed equally.

For numbered affiliations see end of article.

**Correspondence to**
Dr Ekaterina Ivanova;
ekaterina.ivanova@ki.se

## ABSTRACT

**Introduction** Individual cognitive behavioural therapy (CBT) with exposure and response prevention is an effective treatment for obsessive–compulsive disorder (OCD). However, individual CBT is costly and time-consuming, requiring weekly therapy sessions for 3–4 months. A 4-day intensive version of CBT for OCD delivered in group format has been recently developed in Norway (Bergen 4-day treatment, B4DT). B4DT has shown promising results in several uncontrolled and one small, randomised trial, but its non-inferiority to the gold standard treatment has not been established.

**Methods and analysis** This single-blind, randomised controlled trial including 120 patients (60 per arm) will compare B4DT to individual CBT. The primary outcome is the blind assessor-rated Yale-Brown Obsessive Compulsive Scale (Y-BOCS). We hypothesise that B4DT will be non-inferior to gold standard CBT 15 weeks after treatment start. The non-inferiority margin is set at four points on the Y-BOCS. Secondary outcomes include time to treatment response, cost-effectiveness, response and remission rates, drop-out rates and adverse events.

**Ethics and dissemination** This study has been approved by the Swedish Ethical Review Authority. Hypotheses were specified and analysis code published before data collection started. Results from all analyses will be reported in accordance with the Consolidated Standards of Reporting Trials statement for non-pharmacological trials and Consolidated Health Economic Evaluation Reporting Standards irrespective of outcome.

**Trial registration number** NCT05608278.

## STRENGTHS AND LIMITATIONS OF THIS STUDY

⇒ High statistical power to test the primary hypothesis that Bergen 4-day treatment is not more than four points worse on Yale-Brown Obsessive Compulsive Scale than gold standard cognitive–behavioural therapy 15 weeks after treatment start.
⇒ Prespecified methodology and analysis code.
⇒ The blind assessors are blinded to the entire study design.
⇒ Low statistical power to robustly evaluate the secondary outcomes.
⇒ Participants will receive treatment at clinics located in different parts of Stockholm, which may be an obstacle for recruitment and retention.

## INTRODUCTION

Obsessive–compulsive disorder (OCD) is characterised by recurrent uncontrollable (obsessions), and repetitive behaviours (compulsions)[1]. OCD is highly disabling,[2,3] has a lifetime prevalence of 2%–3%[4] and is associated with increased morbidity and mortality,[5,6] academic underachievement[7] and labour-market marginalisation.[8]

Cognitive–behavioural therapy (CBT) with exposure and response prevention (ERP) is a well-established treatment for OCD,[9,10] recommended by international guidelines as a first-line treatment for the disorder.[11] CBT is most commonly delivered in an individual format with weekly sessions during 3–4 months.[12] This means that patients experience a relatively slow road to recovery. If treatment could be delivered in a more intensive way, for a shorter period, without sacrificing clinical efficacy, patients could return to their normal lives earlier. Furthermore, this may potentially result in more cost-effective use of clinical resources.

Researchers in Norway have recently developed an intensive version of CBT for OCD—the Bergen 4-day treatment (B4DT).[13] Results

from uncontrolled trials indicate that approximately 90% of patients reliably improved[14–17] and that 70% were in remission 4 years after treatment.[16] A recent RCT (n=48) showed that B4DT was superior both to a self-help book for OCD and a wait list control.[18] Thus, B4DT is potentially effective and attractive for patients and caregivers alike, but its efficacy and cost-effectiveness have not yet been investigated in a head-to-head comparison with gold standard individual CBT.

In this single-blind, non-inferiority randomised controlled trial, we will evaluate the efficacy of B4DT compared with gold standard, individual CBT. If, as we hypothesise, B4DT is non-inferior to traditional CBT, it could expand the range of available evidence-based options for this patient group.

## Aim and objectives

The primary objective of the study is to evaluate if the intensive B4DT is a non-inferior alternative to gold standard individual CBT for reducing symptom severity in adults with OCD.

We will compare the two groups 15 weeks after treatment start (primary endpoint) with regard to OCD severity. We hypothesise that B4DT will be non-inferior to standard CBT using a non-inferiority margin of 4 points on the Yale-Brown Obsessive Compulsive Scale (Y-BOCS).[19]

Our secondary objectives are:

1. (Primary endpoint) To test whether B4DT is superior to standard CBT.
2. (Primary endpoint) To compare B4DT and gold standard CBT with regard to:
   a. Speed of response.
   b. Rates of response and remission.
   c. Drop-out rate.
   d. Cost-effectiveness.
   e. Adverse events.
      Due to its short and intensive format, we expect B4DT to provide a faster response than gold standard CBT. For secondary objective 2.b–2.e, we have no direct hypotheses. As this study is not sufficiently powered to make confirmatory claims based on the results of these secondary analyses, results will be interpreted as exploratory.
3. Long-term maintenance of gains.

## METHODS
### Study design

We will conduct a single-blind (blinded outcome assessors), randomised (1:1), controlled, parallel-group, non-inferiority trial comparing intensive 4-day CBT (B4DT) and gold standard individual CBT for adults with OCD residing in Sweden. A total of 120 participants will be assessed before treatment start, at weeks 4, 7 and 15 (primary endpoint), and 7 and 16 months after treatment start (follow-ups). The Consolidated Standards of Reporting Trials flow chart of the trial is shown in figure 1. The outcome measures, statistical models and cut-offs, time points of evaluation, and analysis code were published on osf.io (https://osf.io/w5bfp/) before data collection began. The study will follow Good Clinical Practice and all quality and safety aspects will be regularly monitored by an external party, the Karolinska Trial Alliance. The study has been approved by the Swedish Ethical Review Authority (EPM 2022-01713-01). Potential protocol modifications will be described in detail on osf.io, clinicaltrials.org, and—if the modification touches on an ethical issue—an amendment will be submitted to the Swedish EPM.

## Participants

Patients referred to one of the two involved OCD specialist clinics in Stockholm and self-referred patients will be assessed for eligibility by a staff clinician during a full psychiatric assessment. Candidates that live within approximately 1 hour of both clinics and agree to participate irrespective of allocation site will be offered inclusion if they match the inclusion/exclusion criteria (see table 1). Informed consent will either be obtained during a face-to-face clinician visit or on an encrypted study platform, requiring identification and two-step authentication. Once the patient is included in the trial, data on any concurrent treatments will be collected, but those are not prohibited and will not automatically lead to discontinuation of the patient's participation. Karolinska trial alliance is planning a total of seven visits to the study sites, during which they will monitor the presence of signed informed consent, documentation of inclusion/exclusion criteria and randomisation for each participant.

## Randomisation and concealment

The included participants are randomised consecutively to receive either B4DT (delivered at Ångestenheten, Psykiatri Nordväst), or gold standard CBT (delivered at OCD-programmet, Psykiatri Sydväst), regardless of their clinic of origin. A third party, the Karolinska Trial Alliance, has generated the randomisation sequence using the digital system TENALEA that is balanced by being built up of blocks of 4 and 6 occurring in random order. The randomisation is implemented consecutively in TENALEA, which requires an authorised study coordinator to provide a participant ID to generate a unique allocation certificate with participant ID, allocation, personnel ID and time stamp.

Assessors will be blind to the study objectives as well as group assignment up to the last follow-up. If the blinding is broken, the participant will be assigned another assessor at subsequent follow-ups. The assessors will be asked to guess the intervention received at each assessment after randomisation in order to control for quality of the blinding. The following study personnel will remain blind to group allocation until the final data analyses are completed: principal investigator, data managers, primary outcome assessors, trial statistician and health economist.

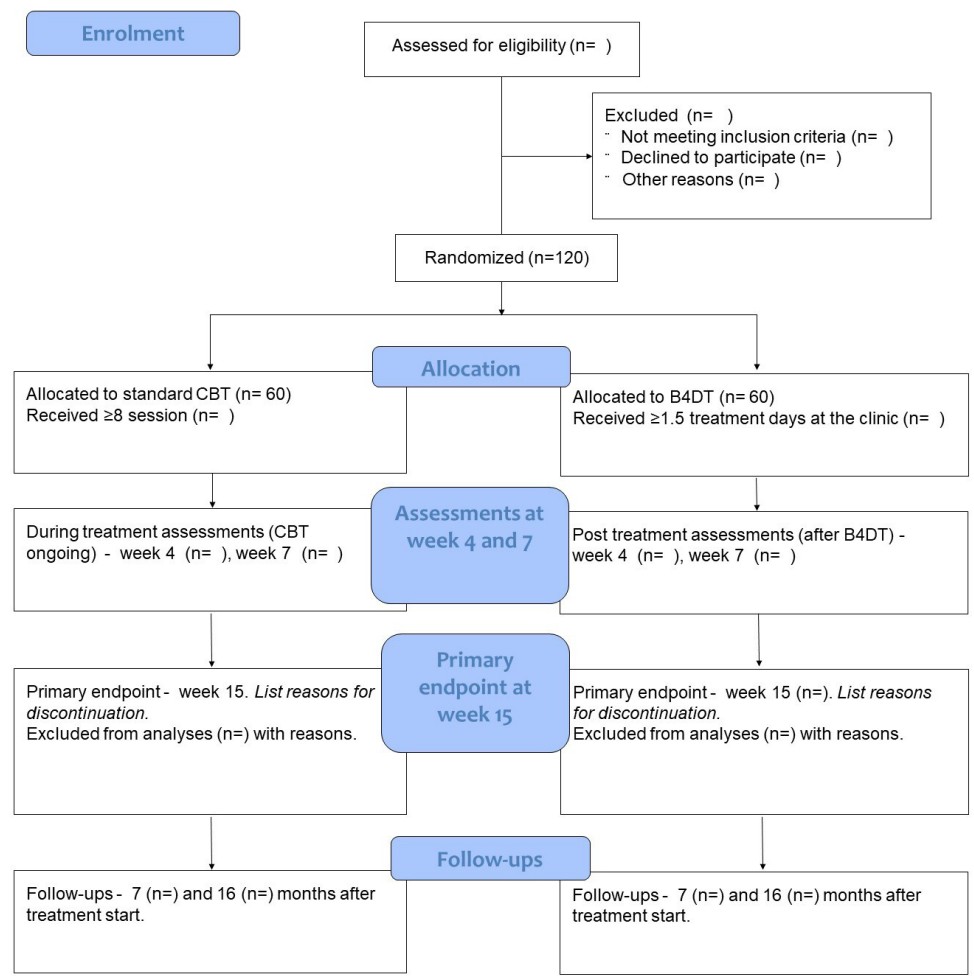

**Figure 1** CONSORT flow diagram. B4DT, Bergen 4-day treatment; CBT, cognitive–behavioural therapy; CONSORT, Consolidated Standards of Reporting Trials.

## Interventions

### Bergen 4-day treatment

Patients in this arm will receive an intensive treatment delivered in a combined individual and group format.[18] The group sizes will be 3–6 participants with a 1:1 patient to therapist ratio. In the week leading up to the intensive part of the treatment, participants will have two scheduled phone/video calls with a therapist to encourage participants to prepare for the upcoming exposure tasks. Day 1 of the intensive treatment includes psychoeducation about the B4DT-adapted rationale for ERP and deciding on exposure tasks. Days 2 and 3 focus on individually tailored and therapist-assisted individual ERP in as many relevant settings as possible (up to 7 hours of ERP per day). In the evenings, patients should continue with self-guided ERP and may receive therapist support via text messages or phone calls on demand. In the afternoon of day 3, patients can invite relatives and friends to a psychoeducation session. Day 4 of the intensive treatment focuses on treatment summary and relapse prevention, as well as planning self-guided ERP for the upcoming 3 weeks. Fifteen weeks after treatment start, participants have individual follow-up sessions at the clinic where they summarise their experiences after completing treatment.

There is no ERP during this session, and it is not a part of the current trial.

All groups will be led by a therapist with expertise in B4DT. All therapists will have participated in the Norwegian OCD training programme[13] or have documented equivalent training. Prior to participation, all therapists will have participated in at least two B4DT groups.

### Individual CBT

Patients will receive 16 weekly 90 min sessions (twice weekly during the first 2 weeks) of individual CBT for OCD with an emphasis on ERP, delivered over 14 weeks according to a validated protocol.[12] Sessions 1–2 contain psychoeducation about OCD and CBT, goal setting and planning of ERP exercises. Sessions 3–14 include therapist-guided ERP (at the clinic, in the patients' homes or elsewhere as needed) with planned self-practice ERP between sessions. Sessions 15–16 contain a summary of the treatment and lessons learnt, as well as relapse prevention and planning of continued self-practice ERP.

In this arm, therapists will be licensed psychologists with extensive experience of treating OCD prior to getting involved in the trial. Trainee psychologists under supervision will not be involved in the treatment of participants.

**Table 1** Overview of inclusion and exclusion criteria

| Inclusion criteria | ≥18 years of age. |
| --- | --- |
| | Primary diagnosis of OCD according to DSM-5. |
| | Clinician-rated Y-BOCS score of ≥16. |
| | Written informed consent. |
| | Being willing and able to attend treatment at any one of the two treatment clinics, regardless of the clinic where the initial assessment took place (the two clinics are located at different locations in Stockholm, about 20 km apart). |
| | Being fluent in Swedish. |
| Exclusion criteria | Other psychological treatment for OCD planned during the trial period. |
| | Completed CBT with exposure and response prevention for OCD in the last 12 months. |
| | Changes in psychotropic medication within the last 2 months. |
| | Bipolar disorder. |
| | Psychosis. |
| | Alcohol or substance dependence. |
| | Organic brain disorder. |
| | Hoarding disorder or OCD with primary hoarding symptoms. |
| | Suicidal ideation that would warrant close monitoring. |

CBT, cognitive–behavioural therapy; DSM-5, The Diagnostic and Statistical Manual of Mental Disorders, Fifth Edition; OCD, obsessive–compulsive disorder; Y-BOCS, Yale-Brown Obsessive Compulsive Scale.

In both arms, inactive patients will be encouraged to engage in the treatment by the clinicians according to the standard clinical routines—such phone calls and letters. Apart from these clinical routines, no extra effort will be made to ensure patients' adherence, to maximise generalisability of the treatment uptake.

### Therapist competence and adherence

Therapists in both arms will be licensed clinical psychologists or psychologists under supervision employed at the two specialist OCD clinics. All sessions will be audiotaped, 20% of the recorded sessions will be randomly selected and rated by independent expert psychologists (two for each treatment) not otherwise involved in the project. Adherence to protocol will be rated using checklists developed specifically for the study and for each intervention. Therapist competence will be rated using the Cognitive Therapy Scale-Revised (CTS-R).[20]

### Sample size calculation

A power calculation was conducted to determine the number of participants needed to test the primary hypothesis that B4DT will be non-inferior compared with gold standard CBT at the prespecified margin of 4 points on the Y-BOCS. There are no specific guidelines on appropriate non-inferiority margins in research on OCD; however, a common rule of thumb is half the placebo-controlled effect of gold standard treatment versus control conditions.[21] The chosen margin of 4 is smaller than half of the 10-point effect that has been demonstrated in ERP trials using Y-BOCS.[10] Importantly, from a clinical perspective, a 4-point effect is small enough to consider other factors than symptom reduction, for example, patient preference, cost-effectiveness and adverse events. Last, this margin gives similar precision compared with previous non-inferiority studies on OCD.[22]

Clinical outcome data from a recent randomised controlled trial[23] and regular clinical practice (B4DT) were obtained to simulate study data. The following parameters were used: A pretreatment mean of 22.6 (SD=3.78) for both arms, and a post-treatment mean of 12.9 (SD=4.07) for the gold standard CBT arm. Statistical power was then assessed by simulating 10 000 datasets assuming different true group differences. An analysis of covariance (ANCOVA) was then conducted on each dataset to test for between-group differences at post-treatment accounting for pretreatment scores. With 60 participants per arm, no difference in efficacy between the treatments, and 10% missing values, this test has >95% power to demonstrate non-inferiority (which was considered supported if the upper bound was less than 4). Details on this calculation, including analysis code, are provided in the online preregistration (https://osf.io/w5bfp/).

### Measurements

Online supplemental table 1 presents the flow of the recruitment and treatment process and lists clinician-rated and self-rated assessments at the different time points.

The primary outcome measure is the blind assessor-rated Y-BOCS, administered at enrolment, at weeks 4, 7 and 15 (primary endpoint), and at the 7-month and 16-month follow-ups.[19] The Y-BOCS is a gold standard instrument for measuring OCD-symptoms having shown excellent interrater reliability (intraclass coefficient of 0.98) and internal consistency (Chronbach's alpha of 0.89)[19] as well as good validity.[24] Blinded assessors will undergo training prior to conducting clinician-rated assessment and practice on videos of OCD case examples. To be allowed to conduct assessments for this project, assessors will have to deviate no more than 1 point on single items, no more than 2 points on the obsession subscale or the compulsion subscale and no more than 4 points on the Y-BOCS total score compared with expert raters at the two clinics. The expert rater scores will be obtained in the following manner: for a corating session, the experienced clinicians working at both clinics will rate the same case recording. The subgroup of the clinicians whose total score lies inside the range between the 25th and 75th percentiles will be identified. The median values on each item in this

subgroup will be considered the expert rater values. Inter-rater reliability on the Y-BOCS will be reported.

Secondary clinician-administered outcome measures are the Clinical Global Impression-Severity and Improvement (CGI-S, CGI-I, blind-rated),[25] the Patient Exposure/Response Prevention Adherence Scale (PEAS),[26] the Structured Clinical Interview for DSM-5 for OCD and related disorders (SCID-5)[27] and the Mini International Neuropsychiatric Interview.[28] SCID-5 and MINI will be administered at enrollment to confirm the OCD diagnosis and assess comorbidities.

Secondary self-rated outcome measures are the Obsessive-Compulsive Inventory-Revised,[29] self-rated Y-BOCS,[30] Montgomery-Åsberg Depression Rating Scale-Self-Rated,[31] Work and Social Adjustment Scale,[32] Assessing Quality of Life 6 Dimensions (AQoL-6D),[33] Credibility/Expectancy Questionnaire,[34] Working Alliance Inventory-Short Form Revised,[35] the Treatment Inventory of Costs in Psychiatric Patients (TIC-P) resource use questionnaire (for economic evaluation)[36] and the Negative Effects Questionnaire (NEQ).[37] Participants will also rate their treatment preference measured on a 7-point Likert-type scale (pretreatment and post-treatment). At the end of each treatment, the therapist will provide information on the number of sessions the patient attended as well as any extra time (additional phone calls, supervision, administrative tasks) spent due to the needs of the specific patient.

Treatment-response will be defined as a ≥35% reduction on the Y-BOCS and a CGI-I score of 1 ('very much improved') or 2 ('much improved'). Remission will be defined as a score of ≤12 on the Y-BOCS and a CGI-S rating of 1 ('normal, not at all ill') or 2 ('borderline mentally ill'). Recovery will be defined as a sustained remission status at the two long-term follow-ups.[38] Participants who complete less than 8 sessions of CBT or attend at less than 1.5 days of B4DT will be considered non-completers, unless the treatment is terminated by the therapist due to early recovery.

The participants who do not engage in treatment or drop out from the trial will be encouraged to take part in all outcome assessments, prioritising the primary outcome measure at the primary end point if the burden of assessments needs to be reduced to increase compliance. In order to assess quality of the data inserted in the electronic case report forms (eCRF), all the eCRF:s will be compared with the source data for 10% of the participants by the quality monitor Karolinska Trial Alliance.

## Safety and adverse events

Information on adverse events including suicidal ideation, and initiation of additional psychiatric care, will be collected at each assessment following inclusion, as well as through the regular contact between participants and therapists during treatment. The incidents will be handled according to the clinics' routines and described on a case-by-case basis. The NEQ with 32 items on incidents and negative effects will be used at the primary

endpoint and follow-ups. Adverse events will be labelled serious if they are life-threatening, result in death or persistent/significant disability/incapacity, require hospitalisation or its prolongation, or is considered medically significant by the principal investigator. All the personal data will be stored in locked archive rooms with personal access (analogue data), or on encrypted servers, requiring VPN-connection and/or two-step authentication (digital data) in accordance with Region Stockholm and Karolinska Institutet's data management guidelines.

## Statistical analysis

Analysis code and a more detailed analysis plan were uploaded to Open Science Framework before any data had been collected (https://osf.io/w5bfp/, under Preregistration—Analysis plan).

### Primary outcome

The primary non-inferiority hypothesis on the clinician-rated Y-BOCS will be evaluated using an ANCOVA of group differences at week 15 with pretreatment score as covariate. It will be based on all treatment completers. This per-protocol approach is chosen to avoid potential problems with the intention-to-treat approach in a non-inferiority study design, which can bias the results towards the alternative hypothesis of non-inferiority.[39] No data imputation will be conducted at this stage. A 95% CI will be used to model the gold standard CBT versus B4DT effect. Non-inferiority is supported if the upper bound is less than 4 (the non-inferiority margin). This testing procedure has the type I error rate controlled at the 2.5% level for both comparisons.

As sensitivity analyses, the same ANCOVA model will be fitted with an intention-to-treat approach, including all randomised participants. Missing data will then be estimated using maximum likelihood after which the models are refitted. The aim of these steps is to address the potential influence of non-random missing data in the primary per-protocol analysis.

### Secondary outcomes

The superiority hypothesis (that B4DT is more effective than standard CBT) will be evaluated using the exact same model used for the main non-inferiority hypothesis, see the Primary outcome section, and superiority will be considered supported if the upper bound of the 95% CI is less than 0.

### Speed of response, rates of response and remission, and drop-out rates

For other secondary outcomes, we will compute 95% CIs of group differences using intention-to-treat approach without comparing the bounds to any predefined margin. Since we focus on pretreatment to post-treatment comparisons, no data imputation methods will be, therefore, used as default (unless stated otherwise). However, if the proportion of missing data exceeds 10%, we will perform maximum likelihood imputations based on demographics and the scores from the pretreatment

assessment. To assess speed of response, responder status in the two treatment arms will be compared at weeks 4 and 7 using generalised linear model where 95% CIs of the OR of responder status will be reported. The same model will be used to compare responder and remitter status in the groups at week 15 to assess rates of response and remission. Proportions of drop-outs in the two treatment groups will be compared using a two-sided two-proportions z-test.

### Cost-effectiveness

We will conduct a within-trial economic evaluation[40] whereby outcome and cost data will be compared at week 15. It will encompass three analyses: (1) a cost-effectiveness analysis using responder status as outcome; (2) a cost-effectiveness analysis using the remitter status as outcome and (3) a cost–utility analysis using the outcome quality-adjusted life-years (QALYs) measured using the AQOL-6D.[33]

Health-related quality of life will be collected using the AQOL-6D,[33 41] a multiattribute utility instrument for use in economic evaluation. Data on resource use will be collected using an adapted version of the self-reported TIC-P questionnaire.[36] Costs will be measured from three perspectives: a healthcare organisation payer perspective (including intervention costs only), a healthcare system perspective (additionally including costs related to the use of medical resources and medication) and a societal perspective (additionally including social care costs and productivity losses for the individual). Costs from each perspective will be analysed in relation to clinical efficacy (responder status; cost-effectiveness analysis) and QALYs (cost–utility analysis). Results will be presented as incremental cost-effectiveness ratios (ICER), as the ratio between the difference in costs and the difference in health outcomes ICERs will be put against values of willingness to pay for a QALY to determine cost-effectiveness.

Standard economic evaluation techniques will be used to explore uncertainty around the cost and effect data, which will be represented on cost-effectiveness planes. The probability of cost-effectiveness according to different willingness-to-pay thresholds will be represented on cost-effectiveness acceptability curves.[42]

### Adverse events

A regression analysis with treatment group as predictor will be used to model the total score from the NEQ at week 15. Additional semistructured questions about severe adverse events will be reported in a frequency table.

### Long-term maintenance of gains

At 7-month and 16-month follow-ups, we will test both the non-inferiority and superiority hypotheses, as well as assess the rates of response, remission and recovery, cost-effectiveness and adverse events. For each outcome measure, we will use the same analysis method as at the primary endpoint described above. The generalised linear model used for rates of response and remission will be applied for rates of recovery.

### Additional outcomes used for quality assessment
### Therapist competence and adherence

Therapist competence (scored using the CTS-R) and protocol adherence will be presented using descriptive information (eg, means and SD), as well as inter-rater reliability between independent assessors.

### Treatment preference, credibility, working alliance and patient adherence

Responses to the question about treatment preference will be presented descriptively at pretreatment and week 15 (means and SD). An exploratory analysis will evaluate whether the degree of agreement between pretreatment preference and treatment received moderates the main Y-BOCS outcome.

Total score means with SD for treatment credibility, working alliance and patients' adherence will be presented for both treatments.

## Patient and public involvement

Patient representatives have not been involved in the design of the current study. The Swedish OCD Foundation supports the study by providing information about the project to their members and the public through their channels. Results from the study will be reported to the public and relevant patient organisations through open access publishing and lectures.

## DISCUSSION

This study is the first trial to directly compare an intensive treatment for OCD, the B4DT with gold standard individual CBT delivered over 14 weeks. The study is well powered to test whether B4DT is non-inferior to gold standard CBT in terms of symptom reduction with a 4-point non-inferiority margin. If the two treatments are comparable in terms of clinical efficacy, B4DT should be considered a candidate for implementation in regular care as the intensive format may be more suitable or preferable for some patients.

## Limitations

First, the two recruitment sites in this study are located in different parts of Stockholm. This means that some participants might prefer to be randomised to the clinic closest to their home, which could pose a risk of selection bias. Second, this study does not have sufficient statistical power to robustly assess the secondary outcomes that will be interpreted as exploratory. Last, therapists and patients will be aware of their allocated treatment arm. However, extensive efforts have been made to improve blinding integrity.

## ETHICS AND DISSEMINATION
## Patient safety

The study has been approved by the Swedish Ethical Review Authority (EPM 2022-01713-01). All patients will

receive either gold standard CBT or B4DT delivered by clinicians who are specialised in each of these forms of treatment delivery. Gold standard CBT is already the firsthand choice for this patient group, and its efficacy and safety have been demonstrated in several studies of high quality.[10] While B4DT has not yet been evaluated in an RCT with an active control group, it is an intensive face-to-face treatment with an emphasis on ERP. Moreover, uncontrolled studies have suggested that the effect is similar, or even larger, compared with gold standard CBT.[15–17] We, therefore, consider the additional risks associated with participating in the B4DT-arm to be small. All participants will be monitored closely during the study and additional treatment will be provided in case of severe deterioration.

Participants referred to the clinics by other healthcare services who turn out not to be eligible will be offered alternative treatment in regular care. Excluded self-referred participants will be guided to adequate care.

## Dissemination

We will adopt practices that are known to increase the trustworthiness of scientific research such as prospectively registering the study hypotheses and the statistical analysis plan, adhering to gold standard outcome reporting, controlling the risk of false positive results by interpreting the secondary analyses as exploratory, and sharing materials needed to evaluate our results (the analysis code, the protocol). A detailed description of the dataset will be saved in a searchable database, SND.[43] The results of the trial will be communicated in open access scientific publications, at scientific conferences as well as meetings with representatives from healthcare system and patient organisations, and in public media.

## Current trial status

Inclusion started in November 2022 and is expected to end in June 2024. At the time of submission, 66 participants had been randomised. The last follow-up appointment is expected to take place in October 2025. Data analysis and reporting of results will begin when all data from the primary endpoint has been collected (October 2024).

**Author affiliations**
[1]Centre for Psychiatry Research, Department of Clinical Neuroscience, Karolinska Institutet & Stockholm Health Care Services, Stockholm, Sweden
[2]Centre for Psychiatry Research, Department of Clinical Neuroscience, Karolinska Institutet, Stockholm, Sweden
[3]Stockholm Health Care Services, Region Stockholm, Stockholm, Sweden
[4]Department of Public Health and Caring Sciences, Uppsala University, Uppsala, Sweden
[5]Division of Psychology, Department of Clinical Neuroscience, Karolinska Institutet, Stockholm, Sweden
[6]Department of Clinical Sciences, Lund University, Lund, Sweden

**Contributors** CR initiated the project and CR, VZI and EI obtained funding for the trial. The study was designed in tight collaboration with DM-C and EA. EI had the overall responsibility for the project's progression. RF and OF developed the preregistered analyses plan and drafted this protocol, RF and EI finished it together.

FS developed the cost-effectiveness analysis plan. SD, SA and MS organised the work at the clinics. All authors collaborated on the proposal and methodology and critically reviewed the final version of this protocol.

**Funding** This study is partly funded by Helse Bergen, the hospital where B4DT was developed. Additional financial support for the study is provided by CIMED (FoUI-960456), Region Stockholm (ALF project RS2020-0731, ALF Postdoc FoUI-963608) and Stiftelsen Söderström-Königska Sjukhemmet (SLS-981976, SLS-980922).

**Competing interests** OF has received speaking fees from the Swedish OCD Association, Insight Events AB and Kry International AB, as well as reimbursement for writing articles for Inside Practice Psychiatry; all outside the submitted work. EI has received speaking fees from the Swedish OCD Association outside the submitted work. Prof Mataix-Cols receives royalties for contributing articles to UpToDate, outside the submitted work. EA and SA receive royalties from Natur & Kultur for books, all outside the submitted work. CR receives royalties from Natur & Kultur, Studentlitteratur and Albert Bonniers Förlag and has received speaking fees from various sources, all outside the submitted work.

**Patient and public involvement** Patients and/or the public were not involved in the design, or conduct, or reporting, or dissemination plans of this research.

**Patient consent for publication** Consent obtained directly from patient(s).

**Provenance and peer review** Not commissioned; externally peer reviewed.

**ORCID iDs**
Ekaterina Ivanova http://orcid.org/0009-0002-6747-5054
Oskar Flygare http://orcid.org/0000-0002-2017-3940
Filipa Sampaio http://orcid.org/0000-0002-5540-9853
Erik Andersson http://orcid.org/0000-0003-0088-8719
Christian Rück http://orcid.org/0000-0002-8742-0168

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
