## [Reviewer comments · BMJ Open]

ARTICLE DETAILS

TITLE (PROVISIONAL)	Study protocol for a single-blind, parallel-group, randomized, controlled non-inferiority trial of four-day intensive versus standard cognitive behavioral therapy for adults with obsessive-compulsive disorder
AUTHORS	Ivanova, Ekaterina; Fondberg, Robin; Flygare, Oskar; Sannemalm, Max; Asplund, Sofia; Dahlén, Sofia; Sampaio, Filipa; Andersson, Erik; Mataix-Cols, David; Ivanov, Volen; Rück, Christian

VERSION 1 – REVIEW

REVIEWER	Jaisoorya, TS NIMHANS, Psychiatry
REVIEW RETURNED	06-Aug-2023

GENERAL COMMENTS	This is excellent clinically relevant/meaningful protocol. The trial has the potential to change practise across the world and bring a significant difference for people with OCD. This can be considered for publication. I have a single question for the investigators. The primary outcome measure is the group differences at 15 weeks which represents the response to treatment for classical CBT while for B4DT (intervention completed in 4 days) the 15-week YBOCS rating will represent a short-term outcome. The comparison of YBOCS a week after completion of B4DT (week 2) with the YBOCS rating one week after completion of CBT (week 15) will possibly represent the immediate response to treatment? If the authors consider the same, then an additional assessment has to be done at week 2, in addition to current protocol. Minor Suggestions  • In addition to Axis 1 disorders, have a diagnosed Personality disorder suggests a poor response to treatment. Screening for personality disorders (say using instruments like SCID-SPQ) and using it as a covariate may help understand reasons for non-response especially to B4DT (suggestion as the initial assessment is very elaborate) • Include as exclusion criteria: Features of Intellectual developmental disorder
--

REVIEWER	Kuroki, Toshihide Kyushu University, Clinical Psychology
REVIEW RETURNED	09-Aug-2023

GENERAL COMMENTS	This study aims to examine the non-inferiority of the effect of a 4-day intensive version of CBT (B4DT) to that of 14-week individual CBT for OCD. The aim of the study is clinically significant, and the
--

	study protocol looks carefully designed, while the following points need further consideration:  1. According to the exclusion criteria noted as “Changes in psychotropic medication within the last 2 months”, the subjects of the study would not exclude patients who have already received maintenance treatment with SSRI but have not improved sufficiently (Y-BOCS score of >16). Would the inclusion of such treatment-resistant OCD cases interfere with the analysis of the outcome? The authors should address the combination with medication treatment. 2. Do neurodevelopmental disorders, including intellectual disabilities, not meet exclusion criteria? 3. Dropout rates: One would expect a lower dropout rate in the group (B4DT) with shorter treatment periods. Would the authors test such a hypothesis?
--	---

REVIEWER	Arumugham, Shyam Sundar National Institute of Mental Health and Neuro Sciences, Psychiatry
REVIEW RETURNED	10-Aug-2023

GENERAL COMMENTS	The study protocol evaluates an important research question comparing the efficacy of the novel intensive 4-day CBT with standard CBT for OCD using a non-inferiority design. Given the high response rates reported in studies evaluating the novel intensive treatment protocol, this RCT plans to conduct a much-anticipated comparison of its efficacy with the standard treatment protocol. It is a well-designed study with adequate sample size, a comprehensive assessment plan and an appropriate analysis plan. The protocol has been described comprehensively, with the analysis discussed in detail and published in OSF. The trial registration details, plan for data storage/analysis and results dissemination have been elaborated adequately. A minor clarification that can be stated explicitly:  • Would additional treatments be allowed following the 4-day treatment during the study period i.e. till the primary endpoint of 15 weeks? For example, change of medications, alternative forms of therapy, neuromodulation etc.
--

VERSION 1 – AUTHOR RESPONSE

Reviewer 1 Dr. TS Jaisoorya, NIMHANS This is excellent clinically relevant/meaningful protocol. The trial has the potential to change practice across the world and bring a significant difference for people with OCD. This can be considered for publication.	
R1 Comment 1	R1 Response 1
I have a single question for the investigators. The primary outcome measure is the group differences at 15 weeks which represents the	We are happy the reviewer made this suggestion. We do already conduct an assessment to evaluate immediate effects of B4DT, it's the assessment at week 4. Week 1 of

response to treatment for classical CBT while for B4DT (intervention completed in 4 days) the 15-week YBOCS rating will represent a short-term outcome. The comparison of YBOCS a week after completion of B4DT (week 2) with the YBOCS rating one week after completion of CBT (week 15) will possibly represent the immediate response to treatment? If the authors consider the same, then an additional assessment has to be done at week 2, in addition to current protocol.	the treatment is the preparation, week 2 is the intensive treatment itself. They are assessed at week 4 using YBOCS that encourages the patients to think about the past week, which is week 3, the week after the treatment (we don't want to ask them about the treatment week itself). However, we will not be comparing the immediate effect of the B4DT with the immediate effect of the standard CBT. By having primary end-point at week 15 for both arms we will make sure that the time parameter is constant across the conditions. Also, it is clinically relevant to know whether the effect of B4DT is maintained at least as long as it takes to provide standard treatment.
R1 Comment 2	R1 Response 2
In addition to Axis 1 disorders, have a diagnosed Personality disorder suggests a poor response to treatment. Screening for personality disorders (say using instruments like SCID-SPQ) and using it as a covariate may help understand reasons for non-response especially to B4DT (suggestion as the initial assessment is very elaborate)	We absolutely agree that it is a very good suggestion. Unfortunately, the trial has already been recruiting since last autumn and we are not screening for personality disorder. However, we will be considering it for the upcoming trials.
R1 Comment 3	R1 Response 3
Include as exclusion criteria: Features of Intellectual developmental disorder	Thank you! We have chosen not to exclude individuals with developmental disabilities to make it easier to generalize our results to regular psychiatric patients. However, the prospective participants are excluded if the assessor deems it very hard for them to follow the study procedures.

Reviewer 2**Dr. Toshihide Kuroki, Kyushu University**

This study aims to examine the non-inferiority of the effect of a 4-day intensive version of CBT (B4DT) to that of 14-week individual CBT for OCD. The aim of the study is clinically significant, and the study protocol looks carefully designed, while the following points need further consideration:

R2 Comment 1

According to the exclusion criteria noted as “Changes in psychotropic medication within the last 2 months”, the subjects of the study would not exclude patients who have already received maintenance treatment with SSRI but have not improved sufficiently (Y-BOCS score of >16). Would the inclusion of such treatment-resistant OCD cases interfere with the analysis of the outcome? The authors should address the combination with medication treatment.

R2 Response 1

Thank you for this comment! We are including patients on stable medication to make our sample match the regular clinical patient population where being medicated with differing degrees of response is common. To include subjects that are medicated has been done in a number of previous trials, and not doing so would affect the ecological validity. We expect the proportion of treatment-resistant cases to be evenly distributed across treatment arms due to randomization. Also, the patients on SSRI are not necessarily treatment resistant as they may have received their SSRI-prescription based on a comorbid condition or as they may have a partial response.

R2 Comment 2

Do neurodevelopmental disorders, including intellectual disabilities, not meet exclusion criteria?

R2 Response 2

We have chosen not to exclude individuals with neurodevelopmental disabilities to make it easier to generalize our results to regular psychiatric patients. However, the prospective participants are excluded if the assessor deems it very hard for them to follow the study procedures.

R2 Comment 3

One would expect a lower dropout rate in the group (B4DT) with shorter treatment periods. Would the authors test such a hypothesis?

R2 Response 3

Thank you! The comparison of drop-out rates is the objective 2.d (included in the originally submitted manuscript).

Reviewer 3**Dr. Shyam Sundar Arumugham, National Institute of Mental Health and Neuro Sciences**

The study protocol evaluates an important research question comparing the efficacy of the novel intensive 4-day CBT with standard CBT for OCD using a non-inferiority design. Given the high response rates reported in studies evaluating the novel intensive treatment protocol, this RCT plans to conduct a much-anticipated comparison of its efficacy with the standard treatment protocol. It is a well-designed study with adequate sample size, a comprehensive assessment plan and an appropriate analysis plan. The protocol has been described comprehensively, with the analysis discussed in detail and published in OSF. The trial registration details, plan for data storage/analysis and results dissemination have been elaborated adequately.

R3 Comment 1

A minor clarification that can be stated explicitly: would additional treatments be allowed following the 4-day treatment during the study period i.e. till the primary endpoint of 15 weeks? For example, change of medications, alternative forms of therapy, neuromodulation etc.

R3 Response 1

We have now added a clarification in the Participants subsection:

“Once the patient is included in the trial, data on any concurrent treatments will be collected, but those are not prohibited and will not automatically lead to discontinuation of the patient’s participation.”

VERSION 2 – REVIEW

REVIEWER	Jaisoorya, TS NIMHANS, Psychiatry
REVIEW RETURNED	28-Oct-2023

GENERAL COMMENTS	The Authors have adequately addressed the comments made and hence the paper can be considered for publication.
--

REVIEWER	Kuroki, Toshihide Kyushu University, Clinical Psychology
REVIEW RETURNED	18-Oct-2023

GENERAL COMMENTS	The manuscript has been revised adequately according to the reviewer's comments. I have no hesitation to accept this paper for publication.
---

VERSION 2 – AUTHOR RESPONSE